# Development of a Versatile Lipid Core for Nanostructured Lipid Carriers (NLCs) Using Design of Experiments (DoE) and Raman Mapping

**DOI:** 10.3390/pharmaceutics16020250

**Published:** 2024-02-08

**Authors:** Carlos Alberto Rios, Roberta Ondei, Márcia Cristina Breitkreitz

**Affiliations:** 1Institute of Chemistry, University of Campinas (UNICAMP), Rua Josué de Castro, s/n, Campinas 13084-971, SP, Brazil; carlosalbertoriosquimicaufla@gmail.com; 2Croda Brazil, R. Croda, 580—Distrito Industrial, Campinas 13054-710, SP, Brazil; roberta.ondei@croda.com

**Keywords:** nanostructured lipid carriers, Raman imaging, design of experiment, classical least squares

## Abstract

The objective of this study was to develop a versatile lipid core for the ‘brick-dust type of drugs’ (poorly water-soluble and poorly lipid-soluble drugs). In the first step, excipients of different polarities were classified according to their behavior in aqueous solutions. Subsequently, binary mixtures were prepared with cetyl palmitate (Crodamol™ CP pharma, Campinas, São Paulo, Brazil) as the solid lipid, and its miscibility with other excipients was evaluated using Raman mapping and classical least squares (CLS). Based on the results, the excipients Crodamol™ CP pharma (hydrophobic), Super Refined™ DMI (dimethyl isosorbide; hydrophilic, Mill Hall, PA, USA), and Super Refined™ Lauryl Lactate (lauryl lactate, medium polarity, Mill Hall, PA, USA) were chosen to compose the lipid core. The ideal proportion of these excipients was determined using a mixture design and the standard deviation (STD) of image histograms as the response variables. After statistical evaluation of the DoE results, the final composition was determined, and drugs with different logP (0 to 10) and physicochemical characteristics were evaluated in the optimized mixture. The drugs butamben (Sigma-Aldrich Co., Spruce Street, St. Louis, MO, USA), tacrolimus (NutriFarm, São Paulo, Brazil), atorvastatin calcium, and resveratrol (Botica da Terra, Campinas, Brazil) presented a homogeneous distribution in the optimized lipid core, indicating that this is a promising system to be used in nanostructured lipid carrier (NLC) formulations of such types of drugs.

## 1. Introduction

Raman spectroscopy is an analytical technique based on the scattering of electromagnetic radiation in the visible to near-infrared region (VIS–NIR) [1]. In Raman spectroscopy, a large volume of data can be generated per measurement; therefore, chemometrics has become a powerful tool for data processing in both industry and universities [2]. Among the areas of chemometrics, design of experiments (DoE) has been highlighted for its ability to identify and quantify the most influential variables. This multivariate approach is crucial for maximizing information collection, observing interactions between factors, and determining the best overall conditions [3,4].

The choice of a suitable design is based on the nature of the variables. If the variables are statistically independent, factorial designs can be used; if many variables are present, a fractional design is indicated. If curvature is identified in the response surface, the use of quadratic models generated by optimization designs, such as central composite (CCD) and Box–Behnken, is indicated [3,4]. In the development of pharmaceutical formulations, the primary goal is often to determine the proportions of the components; therefore, the variables are not independent [4]. In such cases, mixture designs are the best option.

Evaluating the solubility and permeability of the drug is a critical step during the development of new formulations, as both parameters are intimately related to the drug absorption in the body. This forms the basis of the Biopharmaceutics Classification System (BCS), which classifies drugs into four categories: high solubility and high permeability (Class I), low solubility and high permeability (Class II), high solubility and low permeability (Class III), and low solubility and low permeability (Class IV) [5]. Low water solubility limits the development of several pharmaceutical products. This difficulty is more evident in the development of formulations for drugs classified as ‘brick-dust’, which are poorly water-soluble and poorly lipid-soluble [6].

Lipid formulations encompass a broad range of delivery systems, sharing the common use of oils, surfactants, and solvents, and have awakened the interest of the pharmaceutical industry in the last few decades [7,8]. In this scenario, formulations based on a mixture of lipids named solid lipid nanoparticles (SLNs) were initially proposed [9,10]. In SLNs, the lipid core is formed by one or more solid lipids, in which the drug is solubilized. In the second step, a surfactant is added in an aqueous phase under stirring. Even though SLNs have some advantages over other formulations, such as greater in vivo stability and no need for the use of organic solvents, the low amount of drug incorporated into SLNs has impaired their use in many applications [9,10]. Due to these limitations, nanostructured lipid carriers (NLCs) were developed as a new generation of nanoparticles [10,11,12,13]. During the preparation of NLCs, a liquid lipid is incorporated into the solid lipid, solubilizing the drug, making it more stable in relation to crystallization, and allowing the uptake of larger amounts of the drug [12].

In spite of its advantages, the incorporation of ‘brick-dust’-type drugs is still an issue for NLC formulations. This occurs because the liquid lipid present in the core is typically lipophilic and, consequently, does not effectively solubilize the drug. An alternative would be to add a more hydrophilic excipient to the formulation.

Nevertheless, it is highly likely that this excipient will not exhibit miscibility with the solid lipid in the core. Consequently, there is a demand for the development of new formulations of NLCs intended for ‘brick-dust’-type drugs, considering both the solubilization of the drug in the liquid excipient and its miscibility with the solid lipid.

Nanostructured lipid carriers are normally prepared under heating until a visually homogenous mixture is obtained. Nevertheless, miscibility problems can arise after cooling, leading to a further lack of uniformity and phase separation. This is especially important when mixing semi-solid lipophilic and medium polarity or hydrophilic excipients. Chemical images based on spectroscopic techniques (for example, NIR, MIR, and Raman) are important tools for assessing the homogeneity of mixtures [14,15,16] and can indicate heterogeneities at a microscopic level, allowing stability issues to be foreseen.

In addition to the assessment of the distribution of excipients and drugs [17,18,19], chemical images based on vibrational spectroscopy have also been used in the pharmaceutical area to identify adulterations [20], identify and discriminate polymorphisms in drugs [21,22,23], among other applications. To acquire a chemical map, the sample surface is divided into pixels. Subsequently, spectra are collected pixel by pixel, generating a hyperspectral data cube (matrix with dimensions M × N × λ). The first and second dimensions (M and N) correspond to spatial coordinates, while the third contains the spectral profile (λ). After unfolding the data cube into a matrix (MN × λ), it is possible to extract information for the construction of chemical images using univariate or multivariate (chemometric) methods [24,25].

Several chemometric methods can be used to generate chemical images [1,26]. Among the curve resolution methods, the most used are multivariate curve resolution by alternating least squares (MCR-ALS) [27,28,29] and classical least squares (CLS) [25,30]. Both methods follow Equation (1), where the sample spectra matrix is decomposed as the product of a concentration matrix (C), pure spectra (S), and residues (E):(1)$$X={\mathrm{CS}}^{t}+E,$$

In CLS, when the profiles of the pure spectra are known, it is possible to estimate their concentrations in a mixture, assuming that the mixture spectral profile is the weighted sum of pure spectra and the weighting factors are their corresponding concentrations. CLS can be successfully used when the pure spectra are available and no interactions among the components are expected [27].

To interpret chemical image results and assess the homogeneity of the mixture, several authors use the parameters derived from the histogram of predicted concentrations, such as the mean, the standard deviation (STD), the asymmetry, and the kurtosis. However, these parameters describe the dispersion of concentration values but do not indicate the spatial distribution of pixels in the chemical image. Therefore, relying solely on the standard deviation is not sufficient to evaluate homogeneity, as similar STD values may be associated with completely different images. In this situation, Sacré and co-authors proposed a new criterion named the distributional homogeneity index (DHI) [30].

This new parameter is based on the merging of neighboring pixels to form a macropixel (2 × 2, 3 × 3, …) whose concentration value is the average of the values of the grouped pixels [31]. Macropixels move across the image, from the upper to the lower side, line to line, left to right, column to column, and the standard deviation is calculated from the average values obtained for each micropixel. Finally, the deviation values are plotted against the macropixel size, generating a homogeneity curve [32]. In the next step, pixels from the original map are randomly arranged, and a new homogeneity curve is constructed. After finding the area under the curve for both cases, the DHI value is obtained by dividing the area of the original map by the area of the randomized map. The higher the value for DHI, the lower the homogeneity of the mixture.

The aim of this study was to develop and optimize a versatile NLC core, especially for the ‘brick dust type of drugs’. The homogeneity of the mixtures of different excipients with cetyl palmitate (a solid lipid largely used in NLC development) was evaluated using chemical images (CLS), and the ideal composition was determined using a mixture design. The final composition was evaluated by the incorporation of drugs with different logP (0 to 10).

## 2. Materials and Methods

### 2.1. Excipients

Crodamol™ CP pharma (Code: ES68635, Batch: 0001785465, INCI name: Cetyl palmitate, Campinas, São Paulo, Brazil), Super Refined™ Soybean Oil (Code: SR49571, Batch: 0001796977, INCI name: Glycine soja oil, Mill Hall, PA, USA), Super Refined™ Sesame Oil (Code: SR40294, Batch: 0001802159, INCI name: Sesamum indicum seed oil, Mill Hall, PA, USA), Super Refined™ Oleic Acid (Code: SR40211, Batch: 0001778173, INCI name: Oleic acid, Mill Hall, PA, USA), Super Refined™ GTCC (Code: SR71544, Batch: 0001926036, INCI name: Caprylic/capric triglyceride, Shiga, Japan), Croduret™ 40 (Code: ET02113, Batch: 0001833931, INCI name: PEG-40 hydrogenated castor oil, Rawcliffe Bridge, East Riding of Yorkshire, UK), Super Refined™ Lauryl Lactate (Code: SR41856, Batch: 0001842276, INCI name: Lauryl Lactate, Mill Hall, PA, USA), Super Refined™ PEG 400 (Code: SR40377, Batch: 0001747476, INCI name: Polyethylene Glycol 400, Mill Hall, PA, USA), Super Refined™ Propylene Glycol (Code: SR40836, Batch: 0001603745, INCI name: Propylene Glycol, Mill Hall, PA, USA), Super Refined™ DMI (Code: SR40492, Batch: 0001796734, INCI name: Dimethyl isosorbide, Mill Hall, PA, USA), Crodasol™ HS HP (Code: ET40486, Batch: 0001773939, INCI name: PEG 660 12-Hydroxystearate, Mill Hall, PA, USA), Super Refined™ GMCC (Code: SR41366, Batch: OP-474, INCI name: Glyceryl Caprylate/Caprate, Mill Hall, PA, USA) and Super Refined™ PGMC (Batch: 347-04-011, INCI name: Propylene Glycol dicaprylate/dicaprate, Mill Hall, PA, USA).

All excipients were donated by Croda do Brazil (Campinas, SP, Brazil, and headquarters in Snaith, UK). Additionally, all other chemicals and solvents were of analytical grade.

### 2.2. Solubility Tests

The solubility test is a common and essential step in any pre-formulation study. In this test, 1.0 mL of each excipient was added to 60.0 mL of distilled water in a beaker.

After manually stirring, the mixture was permitted to stand undisturbed for 48 h. After this period, a visual assessment of the solution’s appearance was conducted. Excipients were categorized as hydrophobic if there was no interaction with water, i.e., if an oil droplet was formed on the surface of the solution. Excipients were deemed hydrophilic if no aspect of immiscibility (precipitate or haziness) was observed.

In the case of excipients exhibiting intermediate behavior (turbidity or small droplets partially miscible with water), they were classified as medium polarity. It was considered that the excipient has surfactant activity in case of foam formation upon mixing. The procedure was conducted at the laboratory bench under ambient temperature conditions (25 °C).

### 2.3. Tablet Preparation

The excipients used were Super Refined™ Soybean Oil, Super Refined™ Sesame Oil, Super Refined™ Oleic Acid, Super Refined™ GMCC, Super Refined™ GTCC, Super Refined™ DMI, Super Refined™ Lauryl Lactate, Super Refined™ PEG 400, Super Refined™ PGMC, Super Refined™ Propylene Glycol, Croduret™ 40, and Crodasol™ HS HP.

The proportions between Crodamol™ CP pharma and excipients in the binary mixtures were 3:1, 1:1, and 1:3 (*w*/*w*). To prepare the tablets for Raman analysis, a mass of solid lipid (Crodamol™ CP pharma PF 54 °C) was weighed and melted using a heating plate with a magnetic stirrer at 65 °C. (Model MS-H-S, Lab1000, 530 W, 220 V)

Then, a mass of each of the excipients was added with manual stirring. The total mass of the tablet was fixed at 3.0 g. After observing a homogeneous aspect, the mixture was transferred to a metal lid lined with a piece of aluminum foil. After 24 h, the solid mixture was removed from the metal lid in the form of a tablet. An authentic replication of each tablet was prepared.

### 2.4. Raman Spectra

Raman spectra were collected using a full confocal Raman microscope XploRA-ONE (Horiba, Piscataway, NJ, USA) with a two-dimensional CCD detector (−60 °C TE air cooled, 1024 × 256-pixel sensor), 785 nm laser (100–120 mW) and a 10× objective. The spectral region used was 1800–200 cm^−1^ with a spectral resolution of 7.5 cm^−1^. Raman spectra were collected using 1 accumulation with a 3 s exposure time. A central region of the tablet was delimited by 40 × 40 points (total area of 16 mm^2^). The distance between two consecutive mapping measurements was fixed at 100 µm. In total, 1600 spectra were collected for each map. The total mapping time was 90 min.

### 2.5. Chemical Maps

In the collection of spectra, a cube of hyperspectral data with dimensions M × N × λ, with M and N corresponding to spatial information and λ to spectral information, was generated. The first step was to transfer the data from the equipment software (LabSpec v. 6.7.1.10, Horiba) to the MATLAB workspace (Mathworks, Natick, MA, USA, v. 8.3, 2014). Then, the data cube was unfolded into a two-dimensional array of dimensions MN × λ. In the spectra pre-processing step, the asymmetric least-squares algorithm was used (AsLS; λ = 10^5^ and *p* = 0.001) [33], cosmic peaks were removed using the algorithm developed by Sabin et al. (k = 11) [34], and the spectra were normalized by the unitary length vector. Chemical maps were generated using classical least squares (CLS) with the PLS-Toolbox (Eigenvector Research, Manson, WA, USA, v. 8.6.2).

### 2.6. Design of Experiments (DoE)

A mixture design was developed and evaluated by Design Expert^®^ version 11 (Stat-Ease Inc., Minneapolis, MN, USA). The sum of excipients was set to 90% *w*/*w* of the formulation to accommodate a maximum of 10% *w*/*w* of an API in a second step.

The excipients Crodamol™ CP pharma, Super Refined™ DMI, and Super Refined™ Lauryl Lactate were chosen as the independent variables in this step. The range of Crodamol™ CP pharma (lipid solid; X_1_), Super Refined™ DMI (hydrophilic excipient; X_2_), and Super Refined™ Lauryl Lactate (medium polarity, liquid lipid; X_3_) were set to 40–70 (% *w*/*w*), 10–40 (% *w*/*w*), and 10–40 (% *w*/*w*), respectively (Table 1).

The composition of the mixtures was determined by a simplex-centroid mixture design with duplicates at the vertices and three central points. In total, 15 experiments were made. The mass/mass percentages (% *w*/*w*) of each excipient in the mixtures are indicated in Table 2.

### 2.7. Evaluation of the Optimized Mixture Profile Using Different Drugs

As a proof of concept, the following drugs with different partition coefficients (0 < logP < 10) were added to the optimized mixture of excipients (Appendix A): caffeine (Batch: 141124055, Code: 37886, NF: 109143, Botica da Terra, Campinas, SP, Brazil); acetaminophen (Batch: 1602492, Code: 50015, NF: 190482, Botica da Terra, Campinas, SP, Brazil); butamben (Butyl 4-aminobenzoate, CAS: 94-25-7, Batch: 0001452599, Sigma-Aldrich Co., Spruce Street, St. Louis, MO, USA); tacrolimus (Batch: 03513030, Code: 03513030/F1(1), NutriFarm, São Paulo, SP, Brazil); atorvastatin calcium (Batch: ATC20376, Code: 5355, NF: 152593, Botica da Terra, Campinas, SP, Brazil); resveratrol (Batch: 22031001, Code: 8747, NF: 158916, Botica da Terra, Campinas, SP, Brazil); and coenzyme Q10 (Batch: 1122020044, Code: 38702, NF: 722656, Botica da Terra, Campinas, SP, Brazil).

These drugs varied from very hydrophilic to very lipophilic and included ‘brick-dust’ type of drugs. All materials presented an analytical grade of purity (>99%). Authentic replicates of the tablets were prepared following the methodology described in Section 2.3.

## 3. Results and Discussion

### 3.1. Solubility Tests

The initial part of this study involved classifying the excipients based on their behavior in water (hydrophilic, hydrophobic, and medium polarity). A total of 13 excipients were evaluated. The solubility tests were conducted by adding approximately 10.0 mL of each excipient to a beaker of distilled water. After allowing the mixture to rest, excipients were classified according to their behavior as hydrophobic, medium polarity, or hydrophilic (Appendix A).

For those excipients that exhibited partial miscibility in water, a subdivision was made because some presented oil-in-water characteristics (Super Refined™ PGMC and Super Refined™ Lauryl Lactate), while others presented turbidity in the solution (Super Refined™ GMCC). Super Refined™ Lauryl Lactate demonstrated an intermediate behavior; hence, it was considered worthwhile to investigate for a robust NLC core.

### 3.2. Chemical Maps of Binary Mixtures

In the first step, tablets were prepared from binary mixtures between Crodamol™ CP pharma (solid lipid) and excipients with different characteristics (hydrophilic, hydrophobic, and medium polarity) as observed in the solubility test. During the tablet preparation step, only the excipients Crodasol™ HS HP, Croduret™ 40, Super Refined™ Propylene Glycol, and Super Refined™ PEG 400 presented a visual indication of immiscibility.

After preparing the binary mixtures of the excipients, the chemical images and histograms generated were used to investigate the microscopic miscibility. The histograms provided information about the miscibility from the agreement or deviation from the Gaussian format. If the excipient is well distributed, a sharp Gaussian profile with a defined mean value and a small standard deviation is generated. In order to obtain chemical images using CLS, it is essential to provide the spectral profile of the excipients. The Raman spectra of all excipients are presented in Appendix A.

The oily excipients evaluated were the Super Refined™ Sesame Oil, Super Refined™ Soybean Oil, Super Refined™ Crodamol GTCC, and Super Refined™ Oleic Acid. According to the chemical maps and histograms, all excipients were miscible with Crodamol^TM^ CP pharma in all evaluated proportions (1:3, 3:1, and 1:1 *w*/*w*). This result can be explained based on the oily properties of these excipients.

Figure 1 presents the chemical images of the mixture between Crodamol™ CP pharma and Super Refined™ Soybean Oil (1:1 *w*/*w* ratio). Chemical images for Super Refined™ Sesame Oil (Appendix A), Super Refined™ Oleic Acid (Appendix A), and Super Refined™ Crodamol GTCC (Appendix A) at the 1:1 ratio are available in the Appendix A.

The histogram with relative frequencies for each chemical map is generated from the concentrations predicted by the CLS method, pixel by pixel. The maximum and minimum values observed in the color range are associated with the corresponding maximum and minimum concentrations predicted by the CLS method (X-axis of the histogram). Therefore, the color on the scale indicates the intensity of the presence of a specific component in a particular pixel of the chemical map (maximum values correspond to red and minimum values to blue).

Considering all the excipients analyzed in this study, no significant differences were observed between the proportions 1:1, 1:3, and 3:1 (*w*/*w*). Therefore, only the chemical images in the 1:1 ratio will be presented. Additionally, no differences were observed between the replicates in this study. Consequently, the chemical images from a single replicate will be presented.

The next excipients evaluated were Super Refined™ GMCC and Super Refined™ PGMC. In the solubility test, Super Refined™ GMCC demonstrated turbidity in water, while Super Refined™ PGMC formed oil droplets on the water’s surface. Figure 2 presents the chemical images of the mixture between Crodamol™ CP pharma with Super Refined™ PGMC (Figure 2A) and Super Refined™ GMCC (Figure 2B) in the 1:1 ratio.

According to the histograms and images, adequate miscibility was achieved in all proportions, with a slight increase in homogeneity with the addition of liquid excipients.

The next excipients evaluated were Super Refined™ Lauryl Lactate and Super Refined™ DMI. The chemical images of the mixtures of these excipients are presented in Figure 3.

According to the STD of histograms, adequate miscibility of Super Refined™ Lauryl Lactate (Figure 3A) in Crodamol™ CP pharma was achieved in the 1:1 ratio (the same result was obtained for the other proportions). From the solubility test, Super Refined™ Lauryl Lactate presented oily characteristics; however, it also presented an interaction with water. Therefore, it was classified as a medium-polarity excipient.

From the chemical maps and histograms, Super Refined™ DMI (Figure 3B, ratio 1:1) was not fully miscible in Crodamol™ CP pharma in any proportion, even though phase separation did not occur. Based on the solubility test, it was observed that Super Refined™ DMI was a very hydrophilic excipient, which explains the lack of miscibility with Crodamol™ CP pharma. Nevertheless, a mixing test of Super Refined™ Lauryl Lactate and Super Refined™ DMI (both liquids; therefore, chemical images were not required) indicated that they were miscible in all proportions. This information will be used later on to justify the selection of the excipients for the lipid core.

Finally, for the mixtures involving the hydrophilic excipients Super Refined™ PEG 400 (Appendix A) and Super Refined™ Propylene Glycol (Appendix A) with Crodamol™ CP pharma, it was not possible to obtain a tablet in any of the evaluated proportions, due to the formation of two phases in the beaker. Crodamol™ CP pharma remained solid in the upper part, while the other excipient remained liquid in the lower part. This fact can be explained based on the large difference in polarity of the excipients.

For the excipients Crodasol™ HS HP and Croduret™ 40, it was possible to prepare a tablet; however, it was suspected that two phases were formed in the tablet in all proportions analyzed. To confirm this observation, chemical maps of the upper and lower sides in the 1:1 (*w*/*w*) mixture of each excipient were made.

The chemical maps and histograms from the lower and upper sides of the tablet formed by the mixture between Crodasol™ HS HP and Crodamol™ CP pharma are presented in Figure 4A,B. Chemical images for Croduret™ 40 are available in the Appendix A.

Based on the images, it can be seen that the tablet was formed by two parts, with Crodamol™ CP pharma located in the upper part and Crodasol™ HS HP/Croduret™ 40 in the lower part. From the results, it was possible to conclude that Crodasol™ HS HP, Croduret™ 40, Super Refined™ Propylene Glycol, and Super Refined™ PEG 400 were not miscible with Crodamol™ CP pharma in any of the analyzed proportions.

A significant amount of information was obtained from the solubility tests and chemical images. A brief summary is provided below:The excipients that exhibited a hydrophobic profile (Super Refined™ Sesame Oil, Super Refined™ Oleic Acid, Super Refined™ Soybean Oil, and Super Refined™ GTCC) were miscible in all evaluated proportions with Crodamol^TM^ CP pharma.The hydrophilic excipients, Super Refined™ Propylene Glycol and Super Refined™ PEG 400, were not miscible with Crodamol^TM^ CP pharma (it was not possible to obtain a tablet in any proportion tested).Crodasol™ HS HP and Croduret™ 40 allowed the preparation of a tablet; however, chemical imaging revealed that the excipients were localized in different phases (upper/lower sides of the tablets).The excipient Super Refined™ DMI was the only hydrophilic excipient that formed a tablet; nevertheless, the histogram demonstrated a broad distribution of concentrations.Within the group of excipients classified as medium polarity, all excipients provided suitable miscibility with Crodamol™ CP pharma, with slight minor variations at different concentrations. The best overall proportion was 1:1 (Gaussian profile) in all cases.

Super Refined™ Lauryl Lactate is a medium-polarity excipient not widely exploited in the formulation development of NLCs. Considering its miscibility with Crodamol™ CP pharma, it was selected for the next steps. Moreover, as the purpose of this study is to prepare an NLC that is receptive to ‘brick dust type’ of drugs, it was used in association with Super Refined™ DMI, a hydrophilic excipient capable of solubilizing such type of drugs and the only hydrophilic excipient that was miscible to some extend with Crodamol™ CP pharma (no phase separation was observed). Even though Crodamol™ CP pharma and Super Refined™ DMI are not fully miscible, it is expected that Super Refined™ Lauryl Lactate acts as the bridge, serving as a mutual affinity excipient. It should be highlighted that the miscibility of Super Refined™ Lauryl Lactate and Super Refined™ DMI (both liquid) was evaluated in a beaker, in the proportions 1:3, 1:1, and 3:1 (*v*/*v*), and they were completely miscible.

### 3.3. Development of an Optimized Mixture

After defining the core components of the NLCs, the optimal composition of the mixture was determined using a mixture design. The proportions of each excipient in the mixtures, the order of experiment execution, and the design responses (standard deviations of histograms) are presented in Table 3.

After preparing the tablets, Raman spectra were collected in a region of 40 × 40 points, covering an area of 16 mm^2^. The chemical images and histograms with the values of STD, DHI, and kurtosis of the mixtures are presented in Appendix A. Information on the models and the statistical parameters obtained from the analysis of variance (ANOVA) for all responses are presented in Table 4.

Due to the limitations of using STD to evaluate the homogeneity of a mixture, the DHI parameter was proposed [30]. In practice, DHI indicates the degree of heterogeneity of a chemical image because the DHI value increases as the homogeneity of the distribution map decreases. Therefore, the two parameters can be considered complementary because the STD evaluates the dispersion of pixel concentration values (constitutional homogeneity), while DHI evaluates the way in which these pixels are distributed in the chemical image (distributional homogeneity) [31].

The responses used in the mixture design were only the STD values because the models built using the DHI values in the analyzed range did not present significant regression (Figure 5, DHI). This is evident when observing the DHI values (Figure 5, DHI; points A, B, C) for the mixtures with the most different proportions in the mixture design. In this situation, minimal variation was observed between DHI values, which harmed the construction of models for this parameter. This information was observed for all three responses.

To evaluate DHI in the mixture design, it would be necessary to modify the analyzed range of the three independent variables. The values for STD, DHI, and kurtosis for each excipient present in all prepared mixtures were organized in a mixture triangle (Figure 5). The values described for points A, B, and C were obtained by calculating the average of the replicates.

#### 3.3.1. Response Y_1_: STD Crodamol™ CP Pharma

The first response evaluated was the STD of Crodamol™ CP pharma. Based on ANOVA, there was no significant regression, i.e., the mean represents the data well. This becomes evident upon examining the standard deviation values (Figure 5, STD) associated with Y_1_ at points A, B, and C. For the other responses (Y_2_ and Y_3_), significant variations were observed between the STD values. This indicates that, in the analyzed range (40–70% *w*/*w*), Y_1_ had no significant variation in standard deviation, i.e., the width of the histograms was similar. Therefore, any concentration of Crodamol™ CP pharma within this range can be selected, considering the homogeneity of the formulation.

#### 3.3.2. Response Y_2_: STD Super Refined™ DMI

The next response evaluated was STD for Super Refined™ DMI. For this response, a linear model was suggested (coefficient of determination R^2^ of 0.87). Based on ANOVA, the regression was significant, and there was no lack of fit of the model (Figure 5, STD for *Y*_2_). The model equation (Equation (2)) that relates the independent variables to the response is as follows:(2)$$STD\;\left(Y_{2}\right)=+4.65X_{1}+10.7X_{2}+3.21X_{3},$$

The plots of residuals for the model are presented in Appendix A. In the normal plot graph (Appendix A), it was observed that the points lie on the normal line, indicating that the residuals follow a normal distribution.

In the residuals vs. predicted graph (Appendix A), it was possible to observe that the points are randomly distributed, indicating that the residuals for this response are homoscedastic. In the residuals vs. run plot (Appendix A), it was observed that the points follow a random order as functions of the experiment, meaning they are independent. Therefore, it can be concluded that the model residuals follow a normal distribution and are homoscedastic and independent.

#### 3.3.3. Response Y_3_: STD Super Refined™ Lauryl Lactate, Y_3_

The next response evaluated was the STD of Super Refined™ Lauryl Lactate. The linear model was also suggested. Based on ANOVA, the regression was significant, with an R^2^ of 0.72 due to higher random variation (quadratic fit would not improve the results), and there was no lack of fit of the model (Figure 5, STD for *Y*_3_). The model equation (Equation (3)) that relates the independent variables to the response is:(3)$$\mathrm{STD}\;\left(Y_{3}\right)=+5.16X_{1}+3.20X_{2}+7.90X_{3},$$

The plots of residuals for the model are presented in Appendix A. In the same way as the first response, the residuals followed a normal distribution and were homoscedastic and independent.

### 3.4. Surface and Contour Graphs

In a contour graph, it is possible to observe the variation in the response in relation to the composition of the mixture. The model equation provides the predicted value and is associated with a color. The maximum values are found in the red region, and the minimum values in the blue region. The surface graph is obtained after adding the z-axis to the contour graph (2D). No significant regression was obtained for the Y_1_ response. Therefore, the contour and surface graph were not generated.

The contour and the surface graph for Y_2_ (Figure 6A,B) presented a smaller standard deviation in the mixtures where the excipients Crodamol™ CP pharma and Super Refined™ Lauryl Lactate were in greater proportion and Super Refined™ DMI in a smaller proportion. For Y_3_ (Figure 6C,D), a smaller standard deviation was observed in the mixtures where the excipients Crodamol™ CP pharma and Super Refined™ DMI were in greater proportion and Super Refined™ Lauryl Lactate in a smaller proportion. In the surface graph for Y_2_ and Y_3_, the experimental points were close to the surface, indicating that the proposed linear model was appropriate for both responses.

Finally, the actual vs. predicted graph (Figure 6B,D) for Y_2_ and Y_3_ indicates that the experimental points demonstrate good agreement with the linear model proposed by the software for both models.

The contour plots for Y_2_ and Y_3_ presented opposite profiles. This information could be associated with the difference in polarity between Super Refined™ DMI and Super Refined™ Lauryl Lactate and the miscibility of these excipients with Crodamol™ CP pharma, both in the mixture. Furthermore, the profile observed in the contour plots, numerically, can be understood after analyzing the values for STD. By fixing the percentage of Crodamol™ CP pharma in the mixture (Figure 5, STD), for Y_2_, the STD values decreased with the change between points B, F, and C, in that order.

In the same sequence, for Y_3_, an opposite behavior was observed. Therefore, mixtures with a high percentage of Super Refined™ DMI and low Super Refined™ Lauryl Lactate and vice versa did not present an adequate homogeneous distribution due to the higher STD values (Crodamol™ CP pharma fixed at 40 (% *w*/*w*)). The same information can be observed when analyzing the values for the kurtosis.

For Y_2_, an increase in the value of kurtosis between points B, F, and C, in that order, indicates a change from a wider to a thinner histogram. This profile is consistent with the results obtained previously since a thinner histogram indicates a low value for STD.

For Y_3_, a similar behavior was observed, but in the opposite sequence of points (points C, F, and B, in that order). Due to the opposite behavior observed for Y_2_ and Y_3_, the potentially ideal miscibility will be achieved using a mixture with Super Refined™ Lauryl Lactate/Super Refined™ DMI in a 1:1 ratio. This observation is in accordance with the DHI parameter because the mixture with the lowest value for this parameter, i.e., the mixture that presented the most homogeneous chemical image, was precisely in the 1:1 ratio.

Another approach to selecting the best composition would be using the desirability functions [35]. This approach was proposed by Derringer and R. Suich in 1980 as a tool for the optimization of multiple responses. Initially, an individual desirability function (d_i_) is assigned to each response, ranging from 0 to 1 (e.g., maximization/minimization/find a range, reach a target). Then, they are combined in a global desirability function (D), obtained by the geometric mean of the n individual desirability functions (Equation (4)). Then, the goal is to find the set of independent variables to maximize D.
(4)$$D={\left(d_{1}.d_{2}\ldots d_{n}\right)}^{1/n}$$

The responses used in the mixture design were the standard deviations provided by the histogram obtained using CLS; therefore, the goal was to minimize them (Crodamol™ CP pharma was not considered in the calculations since regression was not significant). The range used for the responses and the criteria used to generate the graph for desirability are described in Table 5. The graph for desirability is presented in Figure 7.

According to the desirability graphs, to minimize the responses (STD), it is more appropriate to work in the central part of the mixture triangle. Therefore, the chemical maps and histograms for points A, F, G, H, I, and J were compared and presented a similar homogeneous distribution of the compounds. (Appendix A) Therefore, point F (Appendix A), with a lower amount of Crodamol™ CP (40% *w*/*w*), Super Refined™ DMI, and Super Refined™ Lauryl Lactate in the proportion of 1:1 (25% *w*/*w*: 25% *w*/*w*), was selected to allow for possible higher amounts of drug to be solubilized.

### 3.5. Incorporation of Different Drugs into the Lipid Core

After determining the ideal proportion of excipients, the optimized mixture was evaluated using drugs with different logP and physicochemical characteristics: caffeine (A, logP = −0.07), acetaminophen (B, logP = 0.91), butamben (C, logP = 2.87), resveratrol (D, logP = 3.10), tacrolimus (E, logP = 3.30), atorvastatin calcium (F, logP = 6.36) and coenzyme Q10 (G, logP = 10.00). In the CLS method, chemical maps are generated using the pure spectra of the components; therefore, Raman spectra of the excipients and drugs were collected and are presented in Appendix A. After obtaining the pure profiles, tablets containing the drugs were prepared, and the surfaces were mapped (Figure 8). The complete sets of chemical images and histograms for both replicates for caffeine (Appendix A), acetaminophen (Appendix A), butamben (Appendix A), resveratrol (Appendix A), tacrolimus (Appendix A), atorvastatin calcium (Appendix A), and coenzyme Q10 (Appendix A) are available in the Appendix A.

Among the drugs analyzed, caffeine (logP = −0.07, Figure 8A) and acetaminophen (logP = 0.91, Figure 8B) were the most hydrophilic ones. Based on the chemical images of both drugs, it is possible to observe a heterogeneous distribution on the surface of the tablet. Furthermore, the wide histogram with an undefined mean value indicates that the drugs were not homogeneously distributed. In this study, the two drugs were employed solely as a proof of concept, as they are not formulated as NLCs.

The next drugs evaluated were butamben (logP = 2.87, Figure 8C), resveratrol (logP = 3.1, Figure 8D), tacrolimus (logP = 3.3, Figure 8E), and atorvastatin calcium (logP = 6.36, Figure 8F). From the chemical maps and the format of the histograms, a homogeneous distribution was observed for all these mixtures—resveratrol presented a slightly wider histogram among the mentioned drugs. Therefore, from the histograms generated, it is possible to infer that the drugs were distributed homogeneously; therefore, they are promising candidates to be incorporated into an NLC. It should be highlighted that all these drugs present ‘brick dust’ characteristics.

The last drug evaluated was coenzyme Q10, the most lipophilic one (logP = 10.0, Figure 8G). The chemical images of the drug presented a highly heterogeneous distribution, which was reflected by the format of the histogram.

Finally, the average values of the parameters obtained in the chemical maps and histograms (STD, kurtosis, and DHI) were calculated and are presented in Appendix A. Graphs were generated by plotting logP values against mean STD and DHI values (Appendix A). The logP versus STD plot (Appendix A) presented a “U” shaped profile, indicating that constitutional homogeneity improved (decrease in STD values) with an increase in logP. Furthermore, drugs at the extremes of the logP range had the highest values for STD (acetaminophen, caffeine, and coenzyme Q10).

Appendix A indicates the presence of a proportional relationship between logP and DHI, suggesting that the uniformity of the distribution decreases as the hydrophobic profile of the drug increases. It is important to note that, for caffeine and acetaminophen, the low DHI values may be attributed to the macropixel calculation method. Therefore, a combined evaluation of STD and DHI is recommended.

## 4. Conclusions

The water solubility test carried out at the beginning of this study was a straightforward yet crucial pre-formulation assessment. It provided valuable insights into the behavior of lipid excipients, taking into account their diverse and varying polarities.

Design of experiments (DoE) and chemical imaging allowed determining the optimal composition of the mixture in the lipidic core of NLCs: Crodamol™ CP pharma (40% *w*/*w*), Super Refined™ DMI (25% *w*/*w*), and Super Refined™ Lauryl Lactate (25% *w*/*w*). The inclusion of the medium-polarity excipient Super Refined™ Lauryl Lactate proved to be an innovative strategy for balancing the composition of the lipidic core and enabled the inclusion of a hydrophilic excipient (Super Refined™ DMI) into the core, which was essential for the solubilization of ‘brick dust’ type of drugs.

Results of the last step of the work proved that drugs with intermediate logP values (3.1 < logP < 6.4, butamben, tacrolimus, atorvastatin, and resveratrol) and ‘brick dust’ characteristics presented a homogeneous distribution within the developed NLC core, therefore are suitable to be incorporated in the proposed formulation. Drugs at the extremes of the logP scale (caffeine, acetaminophen, and coenzyme Q10), used as proof of concept, were indeed not miscible in the proposed lipidic core.

## Figures and Tables

**Figure 1 pharmaceutics-16-00250-f001:**
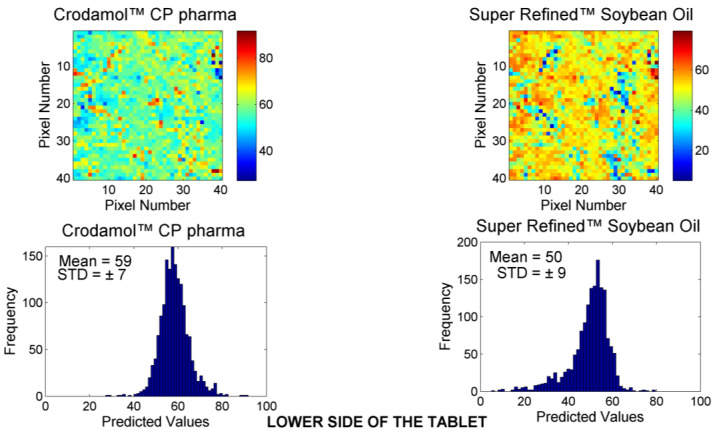
Chemical maps (replicate 1) obtained from the mixture of Crodamol™ CP pharma with Super Refined™ Soybean Oil in the 1:1 *w*/*w* ratio.

**Figure 2 pharmaceutics-16-00250-f002:**
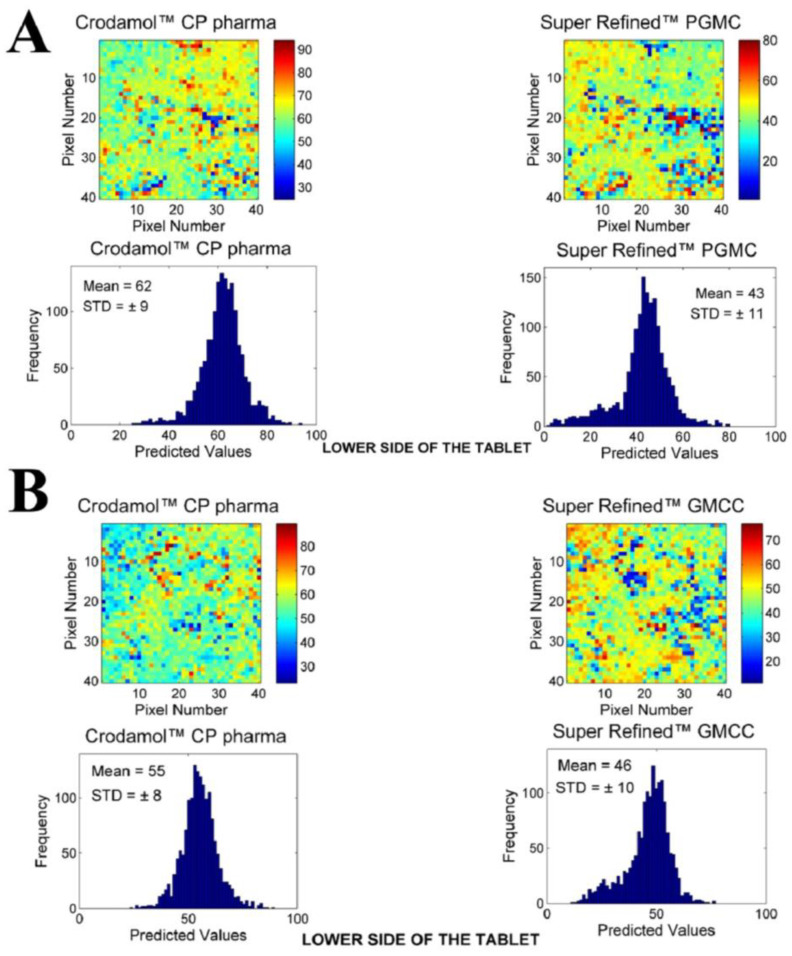
Chemical maps (replicate 1) obtained from the mixture of Crodamol™ CP pharma with (**A**) Super Refined™ PGMC and (**B**) Super Refined™ GMCC in the 1:1 *w*/*w* ratio.

**Figure 3 pharmaceutics-16-00250-f003:**
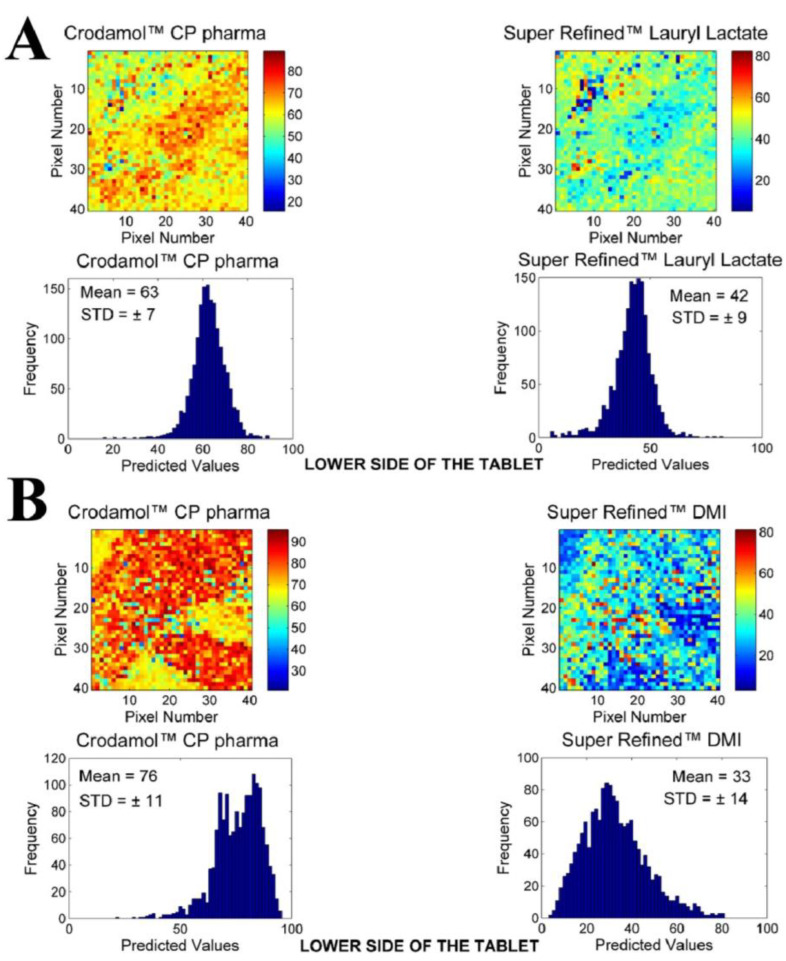
Chemical maps (replicate 1) obtained from the mixture of Crodamol™ CP pharma with (**A**) Super Refined™ Lauryl Lactate and (**B**) Super Refined™ DMI in the 1:1 *w*/*w* ratio.

**Figure 4 pharmaceutics-16-00250-f004:**
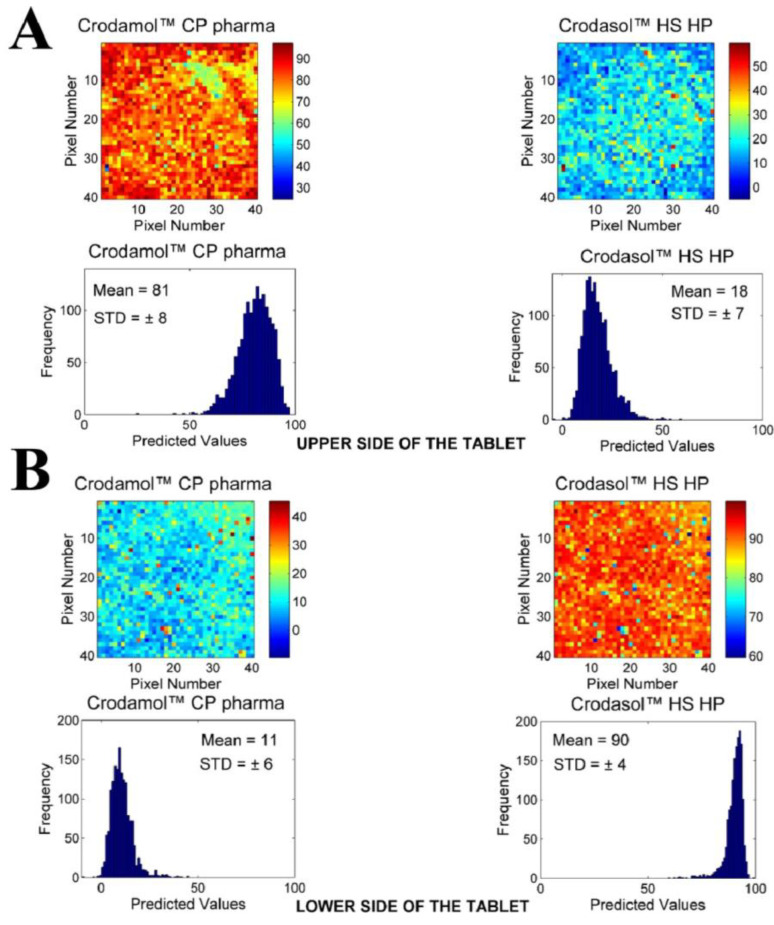
Chemical maps (replicate 1) obtained from (**A**) upper and (**B**) lower side of mixture between Crodamol™ CP pharma and Crodasol™ HS HP (in the 1:1 *w*/*w* ratio).

**Figure 5 pharmaceutics-16-00250-f005:**
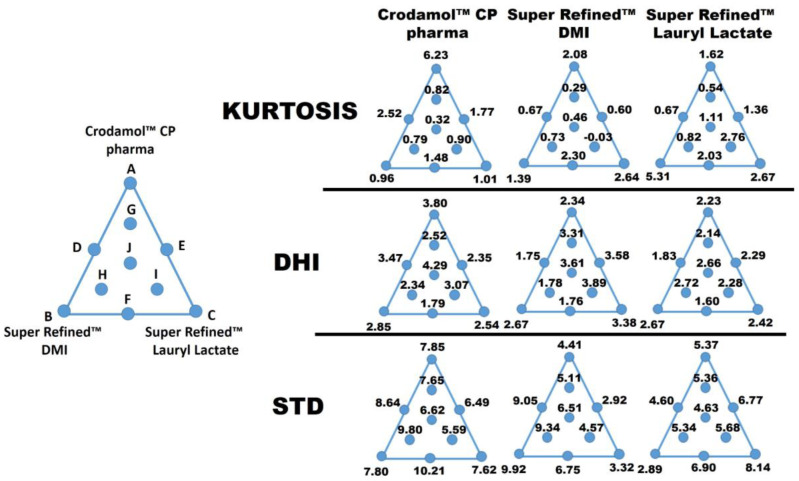
Values of STD, DHI, and kurtosis for each excipient in all prepared mixtures.

**Figure 6 pharmaceutics-16-00250-f006:**
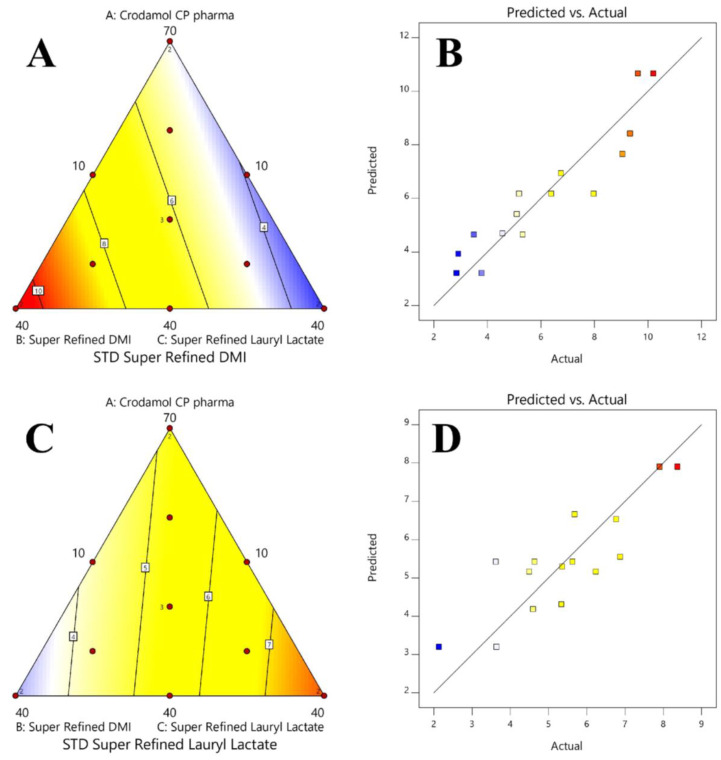
Contour plots and real vs. predicted for responses Y_2_ (**A**,**B**) and Y_3_ (**C**,**D**). The values predicted by the model, in both graphics, are associated with a color scale ranging from blue (minimum) to red (maximum).

**Figure 7 pharmaceutics-16-00250-f007:**
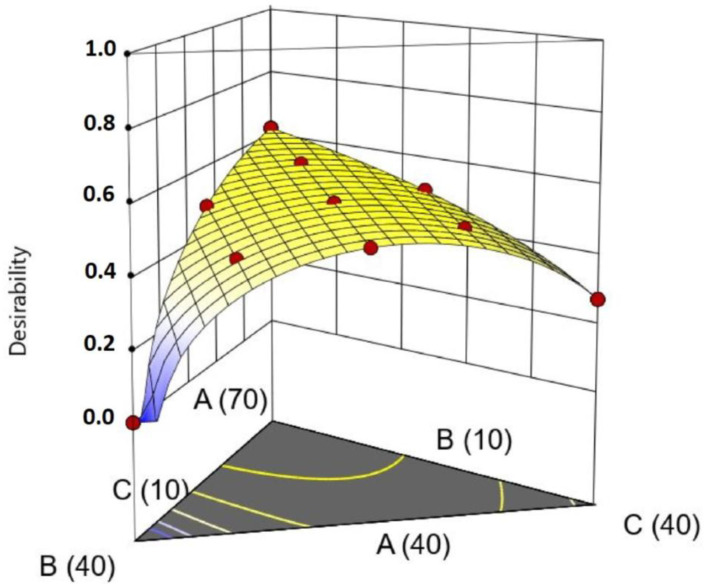
Global desirability graphs: (**A**) Crodamol™ CP pharma, (**B**) Super Refined™ DMI, and (**C**) Super Refined™ Lauryl Lactate. The values for desirability are associated with a color scale ranging from 0 (blue, minimum) to 1 (red, maximum).

**Figure 8 pharmaceutics-16-00250-f008:**
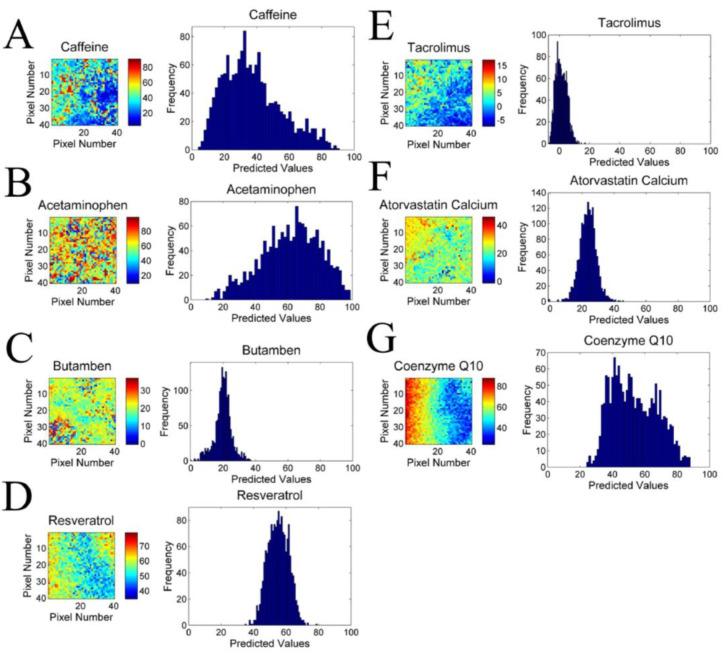
Chemical maps of the mixtures with the excipients Crodamol™ CP pharma, Super Refined™ DMI, and Super Refined™ Lauryl Lactate with the drugs (**A**) caffeine, (**B**) acetaminophen, (**C**) butamben, (**D**) resveratrol, (**E**) tacrolimus, (**F**) atorvastatin calcium, and (**G**) coenzyme Q10.

**Table 1 pharmaceutics-16-00250-t001:** Variables used in the mixture design.

Independent Variables	Range (% *w*/*w*)	
	Minimum	Maximum
X_1_: Lipid solid (Crodamol™ CP pharma)	40	70
X_2_: Hydrophilic excipient (Super Refined™ DMI)	10	40
X_3_: Liquid lipid (Super Refined™ Lauryl Lactate)	10	40
Dependent variables: CLS standard deviation	Target	
Y_1_: STD Crodamol™ CP pharma	Minimize	
Y_2_: STD Super Refined™ DMI	Minimize	
Y_3_: STD Super Refined™ Lauryl Lactate	Minimize	

**Table 2 pharmaceutics-16-00250-t002:** Mass/mass percentages of each excipient in the mixtures.

	Independent Variables		
Mixture	Crodamol™ CP Pharma (X_1_, % *w*/*w*)	Super Refined™ DMI (X_2_, % *w*/*w*)	Super Refined™ Lauryl Lactate (X_3_, % *w*/*w*)
1	70	10	10
2	40	40	10
3	40	10	40
4	55	25	10
5	55	10	25
6	40	25	25
7	60	15	15
8	45	30	15
9	45	15	30
10	50	20	20
11	70	10	10
12	40	40	10
13	40	10	40
14	50	20	20
15	50	20	20

**Table 3 pharmaceutics-16-00250-t003:** Mixture design with proportions and standard deviation responses for each excipient.

		Independent Variables			Dependent Variables (Responses)		
Point	Ord.	Crodamol™ CP Pharma (X_1_, % *w*/*w*)	Super Refined™ DMI (X_2_, % *w*/*w*)	Super Refined™ Lauryl Lactate (X_3_, % *w*/*w*)	STD Crodamol™ CP Pharma (Y_1_)	STD Super Refined™ DMI (Y_2_)	STD Super Refined™ Lauryl Lactate (Y_3_)
A (REP. 1)	3	70	10	10	10.2399	5.3213	6.2387
B (REP. 1)	15	40	40	10	9.7734	10.2075	3.6419
C (REP. 1)	10	40	10	40	8.2503	2.8535	8.3707
D	14	55	25	10	8.6425	9.049	4.5951
E	8	55	10	25	6.4922	2.9182	6.7683
F	2	40	25	25	10.2123	6.7501	6.8751
G	6	60	15	15	7.6515	5.106	5.3594
H	11	45	30	15	9.8017	9.3369	5.3381
I	1	45	15	30	5.5927	4.5703	5.6872
J (REP. 1)	5	50	20	20	5.2051	5.1883	3.6223
A (REP. 2)	4	70	10	10	5.4646	3.5034	4.4966
B (REP. 2)	7	40	40	10	5.8229	9.6252	2.1367
C (REP. 2)	9	40	10	40	6.9807	3.791	7.9058
J (REP. 2)	12	50	20	20	7.209	6.3858	4.6373
J (REP. 3)	13	50	20	20	7.4431	7.9701	5.6311

**Table 4 pharmaceutics-16-00250-t004:** Analysis of variance (ANOVA) for the evaluated responses.

Response	Model	Sequential *p*-Value	SD	R^2^	Adjusted R^2^	Predicted R^2^	
Y_1_	Mean	<0.0001					
Y_2_	Linear	<0.0001	0.988	0.870	0.848	0.799	
Y_3_	Linear	0.000518	0.959	0.717	0.669	0.567	
**Response**	**Source**	**Sum of Squares**	**df**	**Mean square**	**F-value**	***p*-value, prob > F**	**Conclusion**
Y_1_	Model	0	0				
	Residual	44.1	14	3.15			
	Lack of Fit	21.1	9	2.34	0.508	0.822	Not significant
	Pure Error	23	5	4.61			
	Corrected Total	44.1	14				
Y_2_	Model	78.2	2	39.1	40.1	<0.0001	Significant
	Residual	11.7	12	0.975			
	Lack of Fit	5.55	7	0.793	0.644	0.712	Not significant
	Pure Error	6.16	5	1.23			
	Corrected Total	89.9	14				
Y_3_	Model	27.9	2	14	15.2	0.000518	Significant
	Residual	11	12	0.919			
	Lack of Fit	6.26	7	0.894	0.936	0.549	Not significant
	Pure Error	4.78	5	0.955			
	Corrected Total	38.9	14				

Note: df, degrees of freedom; SD (standard deviation).

**Table 5 pharmaceutics-16-00250-t005:** Criteria employed in the overall desirability function.

STD	Lower	Upper	Criteria
Y_1_	5.2051	10.2399	None
Y_2_	2.8535	10.2075	Minimize
Y_3_	2.1367	8.3707	Minimize

## Data Availability

The data used in this study are available to those interested by contacting the author, M.C.B. (marciacb@unicamp.br).

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
