# Peer review of "Development of a Versatile Lipid Core for Nanostructured Lipid Carriers (NLCs) Using Design of Experiments (DoE) and Raman Mapping"

_pharmaceutics, 2024, doi:10.3390/pharmaceutics16020250_

Round 1

Reviewer 1 Report

Comments and Suggestions for Authors

MDPI: pharmaceutics-22822877

The authors investigated lipid mixture using Raman mapping and DoE. The paper is on a topic of importance and will be of interest to others working in the field. I recommend publication with the following changes.

The figures are too small and the texts in the figures are difficult to read.

The authors need to quantify the results of solubility tests in Section 3.1.

Was the DoE presented in Section 3.3 based on a specific drug or just a NLC mixture? X1 + X2 +X3 = 90%. Did the authors have another component in the formula?

The authors used chemical maps and STDs to determine the mixture quality. However, more detailed explanation is required. I recommend that the authors use STD numbers to quantify the mixture quality in section 3.2 and 3.5.

Comments on the Quality of English Language

There are some typos in the MS.

Reviewer 2 Report

Comments and Suggestions for Authors

This is a very interesting manuscript dealing with the application of recent instrumental analytical techniques and statistical approaches for improving the design process of solid lipid nanoparticles by adequate choosing of excipients and proportions. Several drugs of variable polarity were chosen to demonstrate that this novel dosage form is useful for drugs of medium polarity as described by log P values. I think this research could have a great impact in R&D professionals at pharmaceutical industrial level. Thus, in my opinion this manuscript could be accepted after checking some minor points indicated in the attached pdf files. In particular the following:

1. Regarding Supplemental: Please indicate that all these tables and figures are "Supplementary Material" and use pdf format because MS Word file is really heavy (near 900 k).

2. Line 151: Which properties or behaviors of mixtures were considered for this classification?

3. Please improve the quality for clarity of almost all the figures to make them more readable.

4.  Regarding References list: please use italics for journals' names and volumes as required in the respective Guidelines.

5. There are several grammar and typing mistakes as shown that must be checked and corrected.

Comments on the Quality of English Language

Some grammar and typing mistakes have been indicated in the attached pdf file that must be checked.

Reviewer 3 Report

Comments and Suggestions for Authors

Development of a versatile lipid core for Nanostructure Lipid Carriers (NLC) using Design of Experiments (DoE) and Raman mapping.

Employing a mixture design of experiment template coupled with Raman mapping, the authors developed a versatile lipid core for the encapsulation of “brick-dust” active drugs.

 The following corrections are recommended:

1.     Abstract: Authors should add the purpose of the research at the beginning of the abstract.

2.     Line 55 – 59 – Please add more information to this paragraph on class II brick drugs.

3.     Line 75 – avoid the usage of words like won’t. Please write such words in full throughout the manuscript

4.     Line 79 – please write NLC in full at the beginning of the sentence.

5.     Line 148 – Please provide more details on the type of agitation used. Manual or Mechanical using an instrument? If a machine was used, instrument details should be added.

6.     Line 161 – add manufacturer details for hot plate.

7.     Line 204 – Proof of concept

8.     Line 191 – 193 – Why were these three excipients selected for the DOE? Authors should include this in the write-up.

9.     Please include the design template in the methodology i.e. the 15-formulation composition.

10.  1.     Are the NLCs reported herein solid matrices?

 Supplementary Materials:

1.     My understanding is that each NLC contains 10% active drug and 90% excipient total, but this appears to contradict what is depicted in Table S1. Could you please clarify this in the revised submission?

2.     Please improve labels of figure S2 – S8, S11 – S34, specifically font sizes to make legible for readers.

Comments on the Quality of English Language

English language can be improved.

Reviewer 4 Report

Comments and Suggestions for Authors

The manuscript with ID pharmaceutics-2822877 entitled "Development of a versatile lipid core for Nanostructured Lipid Carriers (NLCs) using design of Experiments (DoE) and Raman mapping" is an original research dealing with preformulation studies regarding the NLCs preparation. Raman spectroscopy was use to obtain chemical maps. A Doe with 3 independent variables was constructed resulting in a set of 15 experiments. The data was thoroughly analyzed in terms of DoE and statistical evaluation by applying STD and DHI parameters and comparing them.

Overall, the manuscript is well written and structured dealing with interesting preformulation study in regard to the NLC preparation. Nevertheless, there are several issues which would require the authors attention in order to improve its readability and contribution to the field. They are listed below:

1. Introduction would benefit form better explanation and justification of the research goal. It is not clear what would be the contribution of the study. This is assumption is further supported by the statement in the conclusion "the proposed mixture is not suitable for neither..., as expected." When the results are expected, what is the contribution? Please, state clearly the aim and the novelty.

2. A clear definition and explanation of "Class II brick powder-type drugs" is necessary with proper reference. Currently, the statement "lipophilic, but soluble in organic solvents" is not correct.

3. The authors mention various spectroscopic techniques to assess homogeneity. Please, provide a rationale to chose Raman over other techniques.

4. In Materials section the source of the drugs should be provided in addition to the excipients.

5. How did the authors chose the conditions for solubility test (2.2)? Is it based on any previous work? What are the  criteria based on which the excipients are classified as hydrophobic, hydrophilic and medium polarity? Are they objective?

6. In section 2.6 DoE the drug amount is stated to be set to 10%. Yet, in the supplementary table S1 they vary. Why such difference was present?

7. In the Results and Discussion section very limited references of previous studies regarding the use of the discussed excipients and their miscibility are cited. Please, compare your findings to already published data. Have any of the investigated excipient combinations been studied already - e.g. 10.1208/s12249-011-9733-8; 10.1016/j.ejpb.2007.01.015; or others)? Were the selected model APIs already been incorporated in NLCs (line 562- promising candidates?)

8. Figure 2 is unreadable in its current state. Please, provide better quality.

9. The conclusion section should clearly state the novelty and the scientific contribution of the article. If the results are expected, then why bother to perform the experiment? Furthermore, line 599: "neither for hydrophilic nor too hydrophilic" doesn't actually make sense, please rephrase it.

Comments on the Quality of English Language

The English would require some modifications such as:

1. Too long sentences and the information becomes vague

2. Pronouns are missing in many sentences: e.g. line 36, 53, 104, 233, 235, etc.

3. Some unclear or incorrect sentences: by a one (line 64), is divided several (line 90), line 256, we not miscible (line 326), sing chemical (line 329), bequer (line 349), line 400-401, may be relate the difference (line 471)

4. Some repetitions line 577, line 578

5. In line 379 which is "this parameter" - currently it is not clear to what it is referred. 

Round 2

Reviewer 4 Report

Comments and Suggestions for Authors

The revised version of the manuscript "Development of a versatile lipid core for Nanostructured Lipid Carriers (NLCs) using  Design of Experiments (DoE) and Raman mapping" has addressed all my comments and recommendations. I find it now with increased scientific merit and suitable for publishing in Pharmaceutics.